# Kinematic Parameters That Can Discriminate in Levels of Functionality in the Six-Minute Walk Test in Patients with Heart Failure with a Preserved Ejection Fraction

**DOI:** 10.3390/jcm12010241

**Published:** 2022-12-28

**Authors:** Iván José Fuentes-Abolafio, Manuel Trinidad-Fernández, Adrian Escriche-Escuder, Cristina Roldán-Jiménez, José María Arjona-Caballero, M. Rosa Bernal-López, Michele Ricci, Ricardo Gómez-Huelgas, Luis Miguel Pérez-Belmonte, Antonio Ignacio Cuesta-Vargas

**Affiliations:** 1Grupo de Investigación Clinimetría F-14, Departamento de Fisioterapia, Facultad de Ciencias de la Salud, Universidad de Málaga, 29071 Málaga, Spain; 2Instituto de Investigación Biomédica de Málaga y Plataforma en Nanomedicina (IBIMA), Plataforma Bionand, 29590 Málaga, Spain; 3Departamento de Medicina Interna, Hospital Regional Universitario de Málaga, 29010 Málaga, Spain; 4CIBER Fisio-Patología de la Obesidad y la Nutrición, Instituto de Salud Carlos III, 28029 Madrid, Spain; 5Unidad de Neurofisiología Cognitiva, Centro de Investigaciones Médico Sanitarias (CIMES), Universidad de Málaga (UMA), Campus de Excelencia Internacional (CEI) Andalucía Tech, 29010 Málaga, Spain; 6Centro de Investigación Biomédica en Red Enfermedades Cardiovasculares (CIBERCV), Instituto de Salud Carlos III, 28029 Madrid, Spain; 7School of Clinical Sciences, Faculty of Health, Queensland University of Technology, Brisbane, QLD 4000, Australia

**Keywords:** heart failure, preserved ejection fraction, six-minute walk test, kinematics, physical function, inertial sensor

## Abstract

It is a challenge to manage and assess heart failure with preserved left ventricular ejection fraction (HFpEF) patients. Six-Minute Walk Test (6MWT) is used in this clinical population as a functional test. The objective of the study was to assess gait and kinematic parameters in HFpEF patients during the 6MWT with an inertial sensor and to discriminate patients according to their performance in the 6MWT: (1) walk more or less than 300 m, (2) finish or stop the test, (3) women or men and (4) fallen or did not fall in the last year. A cross-sectional study was performed in patients with HFpEF older than 70 years. 6MWT was carried out in a closed corridor larger than 30 m. Two Shimmer3 inertial sensors were used in the chest and lumbar region. Pure kinematic parameters analysed were angular velocity and linear acceleration in the three axes. Using these data, an algorithm calculated gait kinematic parameters: total distance, lap time, gait speed and step and stride variables. Two analyses were done according to the performance. Student’s *t*-test measured differences between groups and receiver operating characteristic assessed discriminant ability. Seventy patients performed the 6MWT. Step time, step symmetry, stride time and stride symmetry in both analyses showed high AUC values (>0.75). More significant differences in velocity and acceleration in the maximum *Y* axis or vertical movements. Three pure kinematic parameters obtained good discriminant capacity (AUC > 0.75). The new methodology proved differences in gait and pure kinematic parameters that can distinguish two groups according to the performance in the 6MWT and they had discriminant capacity.

## 1. Introduction

Heart failure is not a single pathological diagnosis but a clinical syndrome consisting of cardinal symptoms due to a structural and/or functional abnormality of the heart that may be accompanied by signs and results in elevated intracardiac pressures and/or inadequate cardiac output at rest and/or during exercise [1]. The HF worldwide prevalence ranges from 1% to 3% [2]. Thus, HF is one of the cardiovascular diseases that is increasing in incidence and prevalence due to the ageing of the world population, constituting the leading cause of hospital admissions for people older than 65 and contributing to the increase in medical care costs [2].

Patients with HF have reduced functional aerobic capacity, decreased muscle strength, tiredness, low weekly physical activity, increased time to recover after exercise, fatigue, dyspnea and exercise intolerance [1,3,4]. Furthermore, patients with HF show an impaired physical functional performance, experience a declined ability to carry out their activities of daily living and suffer a reduced quality of life [4]. Falls also are common in patients with HF [5,6]. 40% of patients with HF usually have a fall per year [5,7]. Falls in patients with HF have been associated with a higher body mass index (BMI), poor vision, urinary incontinence, type 2 diabetes, slowness, physical exhaustion, physical functional impairment, and heart failure with a preserved ejection fraction (HFpEF) [5,6,7] Approximately half of the patients with HF present a preserved left ventricular ejection fraction (HFpEF), that is, left ventricular ejection fraction (LVEF) > 50% [1,2]. Patients with HFpEF share the same symptoms as those with HF with reduced ejection fraction (HFrEF), which shows an LVEF < 45% [1]. However, research and clinical experience show that older adults with HFpEF are patients with more significant functional impairment and complex handling than older adults with HFrEF because no pharmacological treatment has demonstrated any clear prognostic benefits [1,8]. This way, older adults with HFpEF also showed reduced muscle strength, limited functional aerobic capacity, slowed gait speed and poor physical function in previous studies [9,10].

Functional symptoms, like accentuated muscle dysfunction, appear in patients with HFpEF and not in patients with HFrEF [11]. Functional aerobic capacity has been inversely correlated to the severity of HF and directly correlated to the prognosis and life expectancy [12]. In addition, maximal oxygen uptake (VO2 max) obtained from a cardiopulmonary exercise test is the gold standard for measuring aerobic exercise capacity and intolerance [13]. Another option for assessing functional aerobic capacity is the 6-minute walking test (6MWT) [14,15]. The 6MWT can predict the prognosis of older adults with HF based on the walked distance [15,16], and the 6MWT distance also correlates with the VO2 max in older adults with chronic HF who do not walk more than 490 meters (m) [17]. Patients with HF who walk a distance lower than 300 m in the 6MWT have a larger risk of hospitalization or mortality [16]. However, previous studies showed that many older adults could not complete the 6MWT due to limited exercise capacity, fatigue and other symptoms such as dyspnea [9,18]. 

Regarding the gait parameters, there are discrepancies in which ones are useful to assess limitations in functional aerobic capacity in HF [19]. Kinematic assessment during the 6MWT has been widely studied [20], but a kinematic analysis during the 6MWT has not been performed in patients with HFpEF. Kinematic assessment could allow quantifying normal and pathological movements, quantifying the degree of functional impairment, planning rehabilitation strategies and evaluating the effect of various interventions [21]. Kinematic parameters also help to identify gait parameters related to limitations in aerobic capacity in HF [19]. In this sense, the acceleration of the trunk in the Y axis has made it possible to differentiate between frail and non-frail older adults in functional tasks [22,23]. The kinematic parameters also make it possible to discriminate between patients with mild cognitive impairment and cognitively healthy older adults [24], patients with different severity of low back pain [25], or between fallers and non-fallers with Parkinson’s disease [26]. Kinematic parameters could even predict falls in older adults [27]. Kinematic parameters may help discriminate and stratify patients with HF based on different functional impairment levels. Thus, the leading aims of the present study are (1) to assess kinematic parameters which could discriminate between groups able to walk more than 300 m in the 6MWT and unable to walk more than 300 m and (2) to assess kinematic parameters which could discriminate between patients with HFpEF who can finish the 6MWT and patients with HFpEF who have to stop during the 6MWT. Secondary aims are (1) to assess kinematic parameters which could discriminate between women and men in the 6MWT and (2) to assess kinematic parameters during the 6MWT which could dis-criminate between patients with HFpEF who had fallen in the last year and those who did not fall in the last year.

## 2. Methods

### 2.1. Design

A cross-sectional study was carried out between April 2019 and March 2020 in the Heart Failure Unit of the Internal Medicine Department at the Regional University Hospital of Malaga (Malaga, Spain). The study was registered on the ClinicalTrial.gov database as NCT03909919.

### 2.2. Participants

The study included the participants according to the following criteria: (1) patients diagnosed with HFpEF according to the European Society of Cardiology Consensus Statement [3]; (2) older than 70 years of both genders. Patients were excluded if they were HFpEF patients with a New York Heart Association (NYHA) class = 4, patients hospitalised 3 months ago or less, patients with a Mini-Mental State Examination (MMSE) score < 24 or patients who were unable to walk or stand up from a chair.

### 2.3. Ethical Issues

Ethical approval was obtained from the Provincial Ethics Committee of Malaga, Malaga, Spain (26032020). The study was carried out following the Helsinki Declaration [28], and the Good Clinical Practice guidelines. The study was implemented and reported according to the Strengthening the Reporting of Observational Studies in Epidemiology (STROBE) Statement [29]. Moreover, all participants in this study were recruited as volunteers and signed an informed consent form before enrolment. They also could leave the study freely at any time.

### 2.4. Outcomes

The Six-Minute Walk Test (6MWT) was carried out in a closed corridor. Two marks were placed on the ground at 30 m, and patients walked from one end to the other for 6 minutes (min). Patients were instructed to walk as quickly as possible and informed of the time elapsed on each lap. The distance from which patients walked for 6 min was recorded [30]. While performing the 6MWT, participants also wore two inertial sensors, one on the back at the L3–L4 level and the other on the sternum, to assess kinematic parameters (Figure 1). It was recorded who did not finish the test and why the patients stopped.

Kinematic parameters were collected with two Shimmer3 (Shimmer Research Ltd., Dublin, Ireland) inertial measurement units (IMUs). Kinematic parameters such as the trunk’s angular velocity (°/s) and linear acceleration (m/s^2^) were assessed using the gyroscope and the accelerometer included in the Shimmer3 IMUs. The Shimmer3 IMU can measure kinematically along three orthogonal axes (X, Y, Z). A sensor was placed snugly secured to the lower back, specifically at the L3–L4 level, and fixed with adhesive tape to reduce error due to the movement of the sensor on the skin. The other sensor was placed in the same way but in the medial third of the sternum in the dorsal area (Figure 1). The three orthogonal axes embedded in the body represented three directional movements: X, lateral movement; Y, vertical movement; and Z, anterior-posterior movement. The Shimmer3 data-sampling rate was set to 256 Hz. 

### 2.5. Procedure

An internal medicine physician (the author LMP-B) recruited and assessed the eligibility of the participants during consultation. If participants met the eligibility criteria, they would be invited to participate in this study. The measurement was all done on the same day. While performing the 6MWT, participants also wore two inertial sensors, one on the back at the L3–L4 level and the other on the sternum, to assess kinematic parameters (Figure 1). Patients were instructed on how to perform the 6MWT. Patients were also instructed to walk as quickly as possible and informed of the time elapsed on each lap [30].

### 2.6. Kinematic Data Processing

The software used to assess and record kinematic parameters was Consensys v1.6 (Shimmer Research Ltd., Dublin, Ireland), a specific and paid software developed for kinematic analysis with Shimmer3. This software allows users to assess and record kinematic parameters, create a CSV file with recorded data, and export it. After each trial, the created CSV were processed and analysed using MATLAB software (Version R2018b, MathWorks, Natick, MA, USA). An own MATLAB code was created specifically for this project to process and analyse the kinematic parameters. The first function of this code is to allow the researcher to select the beginning and the end of the functional test. Once the beginning and the end of the 6MWT are selected, the code analyses the angular velocity and the linear acceleration of both Shimmer3 IMUs, and the code calculates the following gait kinematic parameters from the Y-axis linear acceleration of the shimmer located at the L3–L4 level: total 6MWT distance (m), lap time (s), gait speed (m/s), number of steps and strides (n), step and stride time (s), step and stride length (m), step and stride velocity (m/s), step and stride cadence (steps or strides/min), and finally, step and stride symmetry ratio (higher step or stride time/lower step or stride time). Additionally, the pure kinematic parameters of both inertial sensors were extracted in the three dimensions: mean, maximum, and minimum peak. The interface of the processing code was shown in Appendix A.

### 2.7. Data Analysis

Descriptive, inferential and discriminant analyses were carried out. An absolute frequency and a percentage are described as qualitative measures. Quantitative measures were reported using the mean, the standard deviation (SD), the maximum, and the minimum. Distribution and normality were determined by one-sample Kolmogorov-Smirnov tests (significance < 0.05). The Student’s *t*-test (*t*-test) was used to measure kinematic differences between patients with HFpEF who walked more than 300 m and those patients who could not walk more than 300 m. The *t*-test assessed the kinematic differences between patients with HFpEF who stopped during the 6MWT and those who did not stop. This test was also used to assess the kinematic differences between women and men in the 6MWT, and between patients who had fallen in the last year and those who did not fall. 95% confidence intervals (95%CI) of the kinematic differences were also reported. Levene’s test assessed the variance heterogeneity (significance < 0.15). Four discriminant analyses using a receiver operating characteristic (ROC) curve were performed to assess the discriminant ability of kinematic parameters: (1) between patients who walked less than 300 m [16], (2) people that could not finish the 6MWT, (3) women and men, and (4) patients who had fallen in the last year. The closer the area under the ROC curve (AUC) is to 1, the stronger the relationship [31]. The kinematic parameters that showed an AUC higher than 0.75 were reported in the present study. Lap time and gait speed were not included in the ROC curves that discriminated between patients who walked more than 300 m and patients who walked less than 300 m because they are derived from the total 6MWT distance outcome. A *p*-value of *p* < 0.05 was considered to be statistically significant. All statistical analyses were conducted using the Statistical Package for the Social Sciences (SPSS) 22.0 (IBM, Armonk, NY, USA) for Windows.

## 3. Results

### 3.1. Patients’ Characteristics

Seventy patients with HFpEF were voluntarily included in the study. Participants’ descriptive, anthropometric and clinical variables, as well as blood and urinary biomarkers, are shown in Appendix A. The descriptive kinematic parameters of the complete sample were also presented in Appendix A. The mean age of included patients with HFpEF was 80.74 years old, and the mean LVEF was 60.71. 41 included patients with HFpEF (58.60%) were women. Most patients with HFpEF showed an overweight (40.00%), and 39 patients (55.70%) had fallen in the past 12 months. Most patients with HFpEF showed a NYHA class of II (68.60%). Moreover, included patients showed an average of 8.34 comorbidities and took an average of 10.09 drugs every day. Patients with HFpEF walked an average of 246.21 m (93.33) in the 6MWT, but only 58.60% of patients with HFpEF could finish the 6MWT. The leading cause of stopping the 6MWT was intolerant dyspnea (62.10% of patients with HFpEF who could not finish the 6MWT).

### 3.2. Outcomes from the Study

Lumbar and chest linear pure kinematic analysis with acceleration and angular velocity in the three dimensions during the 6MWT were shown in Table 1. Kinematic gait parameters differences between the ≥300 m and <300 m groups in the 6MWT and between patients who completed the 6MWT and patients who could not finish the 6MWT were shown in Table 2. Pure kinematic outcomes differences in the three dimensions between the ≥300 m and <300 m groups in the 6MWT were reported in Table 3, while these differences between patients who completed the 6MWT and patients who could not finish the 6MWT were reported in Table 4. Kinematic gait parameters differences between women and men in the 6MWT and between patients who had fallen in the last year and those who did not were shown in Table 5**.** Pure kinematic outcomes differences in the three dimensions between women and men in the 6MWT were reported in Table 6, while these differences between patients who had fallen in the last year and those who did not fall were reported in Table 7. The best discriminated kinematic parameters between patients who walked more than 300 m and patients who walked less than 300 m in the 6MWT, between patients who completed the 6MWT and patients who could not finish the 6MWT, between women and men, and between patients who had fallen in the last year and those who did not fall were reported in Table 8.

## 4. Discussion

The objective of this study was achieved, and a new methodology could be presented to analyze the 6MWT test with inertial sensors in patients with HFpEF, which can help to stratify patients with HF based on the functional performance according to the impairment level. There was a difference between groups in all gait kinematic outcomes besides seven parameters with high discriminant values in both classifications.

### 4.1. 6MWT and Gait Speed

Patients with HFpEF who walked more than 300 m showed a higher gait speed (1.0 m/s) than patients with HFpEF who walked less than 300 m (0.54 m/s). Similar results in the other analysis between people who completed the test (0.80 m/s) and people who did not (0.51 m/s). Our results also showed that patients with HFpEF who could not finish the 6MWT walked 105.10 m less than patients with HFpEF who finished the test. 

The mechanisms that walking gets slowed down have had clinical and functional relevance [19], so their study can lead to a better evaluation of HFpEF. Moreover, the average distance travelled in this study (246.21 m) is lower than what was established (300 m) for the risk of death or hospitalization, according to Fuentes-Abolafio et al. [16]. This difference may be influenced by many parameters that we still do not know due to the difficult definition of the phenotype [32], the load of comorbidities [33], other psychological factors [34] or the different fatigue levels. Regarding the latter, fatigue is significant since these patients are more susceptible to central and peripheral fatigue [35]. A distance of 300 m is taken as a symptom of fatigue in the diagnosis of HF if less than it is travelled [36]. It has also been seen that not exceeding it is associated with worse cardiovascular results [37]. More in-depth studies to measure fatigue during the test could show a biomarker that helps define this subtype.

Another aspect associated with worse results is the physical capacity of the subjects. Patients with HFpEF have worse muscle function and more atrophy than HFrEF [11]. Regarding fat tissue, an increase in skeletal muscle fat and less exercise tolerance have been seen in HFpEF [38]. 40% of the patients in the study were overweight, and, looking at the minimum data on the distance travelled (105 m), it is very likely that some of them will have severe cardiovascular problems in the future. These capacities and the level of physical activity of the patients can modify the results of walking, as has been found in the previous studies [39]. Each patient’s baseline physical activity level should be considered in the assessment, together with a more intense neuromuscular analysis, which should include muscle strength, quality and quantity.

The relationship between 6WMT and gait speed is very close because both have similar abilities to predict mortality prognosis in cardiovascular diseases [40], so gait speed is essential. Gait speed was not included in the ROC curves analyses because the total 6MWT distance derivated the gait speed, and the AUC was 1. According to Kamiya et al. [40], gait speed is a good predictor of 6MWT with a test distance of less than 400 m. The average distance in this work was 246.21 m, for which the correlation between both parameters is robust in this study. Compared with another study on mild cognitive impairment reported differences in the gait speed with healthy people [24]. Nevertheless, Panizzolo et al. [19] did not find differences in the gait speed in chronic HF. Future studies need to create more comparisons with healthy people and other subgroups of HF.

### 4.2. Gait Kinematic Parameters

The step time, step symmetry, stride time and stride symmetry were the gait kinematic parameters that best discriminated in the two-discriminant analysis: (1) between patients with HFpEF who walked more or less than 300 m in the 6MWT and (2) patients with HFpEF who could not complete the 6MWT. These facts per the previous insight about the contribution of a short-stepping gait in the limitation of exercise capacity in patients with HF and the different responses between healthy subjects [41]. The exercise capacity and walking cannot be underestimated in HFpEF because it was proved that it is a good indicator of daily life functionality [42], so the type of walking and the objective analysis could give more precise assessments in the future.

All the gait parameters were significantly different regarding the analysis of the variables between the two groups previously mentioned. More differences in cadence and steps/breath between patients with HF and healthy subjects were shown by Clark et al. [43]. However, Panizzolo et al. [19] did not find differences in the stance time, swing time or stride length between patients with HF and healthy subjects. A previous study reported differences in stride length, stride time, stride velocity, swing time, stride width, stance time, single support time and double support time between patients with mild cognitive impairment and cognitively healthy older adults [24]. Similar results were found between fallers and non-fallers with Parkinson’s disease in stride time, stride length and gait speed [26]. Thus, all these results help to affirm that the gait kinematic parameters could be useful to discriminate in patients with HFpEF similar to other clinical populations.

Finally, only stride length was lower in women than in men. This stride length has been associated with the severity of HF, so the women included could have a more severe pathology [41]. No kinematic gait variable discriminated between those who fell and those who had not fallen in the past year.

### 4.3. Pure Kinematics Parameters

More pure kinematics differences were found in people that completed the test in ≥300 m and <300 m. The maximum acceleration and velocity in Y always differed significantly in all the analyses. Although these variables have not been different between patients with chronic heart failure and healthy people, it is known that there are important factors in biomechanics and kinematics that can classify them [19]. The acceleration results reflected interesting data that must be considered to stratify patients. It was noteworthy that the acceleration in the Y axis (9.15 m/s^2^) that would correspond to the vertical movement was greater than the acceleration in the Z axis (3.28 m/s^2^), which corresponds to the person’s anteroposterior movement. If the movement is walking forward, more acceleration can be expected in the Z axis than in the Y axis, but this may be important to elicit future kinematic biomarkers that facilitate assessment. This fact agrees with the study by Galan-Mercant et al. [22] on frail patients in the Timed Up and Go using inertial sensors for kinematics. During the gait phases, the Y axis obtained better results than the Z axis; even the non-frail patients had a more significant difference between these parameters than the non-frail ones [22]. Regarding the ROC analysis, no acceleration parameters obtained a high AUC value. Our results disagreed with Hasnain et al. 2022 because they found one suitable acceleration parameter in the spine [44]. In contrast to these data, a previous study found several parameters trying to distinguish frailty and non-frailty people [45]. A possible reason for this heterogeneity could be further analysis or group characteristics. 

On the other hand, angular velocity results have also been helpful in other pathologies, such as LBP [46]. Three minimum velocity parameters from lumbar and chest movement in the Y and Z axis only showed high results (AUC > 0.75). The importance of this parameter must be well analyzed because it separately focuses on the trunk’s angular movement. When walking, there is not usually much displacement of it. These deviations in the minimum velocity could be attributed to abnormal body posture. Special attention has been required in the lumbar minimum Y movement because it was also significantly different in the group’s comparison. The lack of more relevant discriminant parameters agrees with the results of Hasnain et al. 2022, where no velocity results in the trunk could distinguish the level of physical activity and the appearance of unexpected healthcare encounters in oncology patients with aggressive chemotherapy during walking tests [44]. More studies are needed to explain the possible discriminant capacity of these variables because there are findings supported by previous kinematics studies in other clinical populations with low function capacity [44,45].

### 4.4. Strengths and Limitations

Regarding the performance of the 6MWT, only 58.60% of patients with HFpEF could finish the test. Older adults with HFpEF with limited functional aerobic capacity have difficulty completing the test due to fatigue and other symptoms such as dyspnea. According to the presented results, kinematic assessment is another option that involves fewer space-time resources than the 6MWT. It allows clinicians to stratify their patients functionally.

In addition, this large sample size of recruited patients with HFpEF could reduce any risk of selection or performance bias. The cross-sectional design could reduce any risk of detection and attrition bias as it reduces the possibility of missing data in outcomes. The results with statistically significant differences and statistically non-significant results were presented in this study to avoid publication bias.

Although these strengths, this study presented some limitations. One of these limitations was that the results were not compared to an HFrEF or healthy control group. An analysis of these gait and kinematic variables with other physiological outcomes during the 6MWT could be recommended to complete the clinical assessment. It will consider outcomes such as oxygen consumption, perceived exertion or heart rate because these variables correlated during walking [3]. Thus, this study presented this analysis and methodology, but the results obtained still need to be explored further. Future studies need to relate these movement-based results extracted with this methodology to exercise-related variables and compare them with a similar HF group.

## 5. Conclusions

The results of this study presented descriptive data on the development of the 6MWT test in HFpEF with an inertial sensor-based methodology. Gait kinematic outcomes can be useful to assess HFpEF patients during a different performance of 6MWT. Regarding the pure kinematics results, higher accelerations were observed more times in the vertical movement. Future studies should relate these results with other parameters and look for the stratification of this subtype among the other HF.

## Figures and Tables

**Figure 1 jcm-12-00241-f001:**
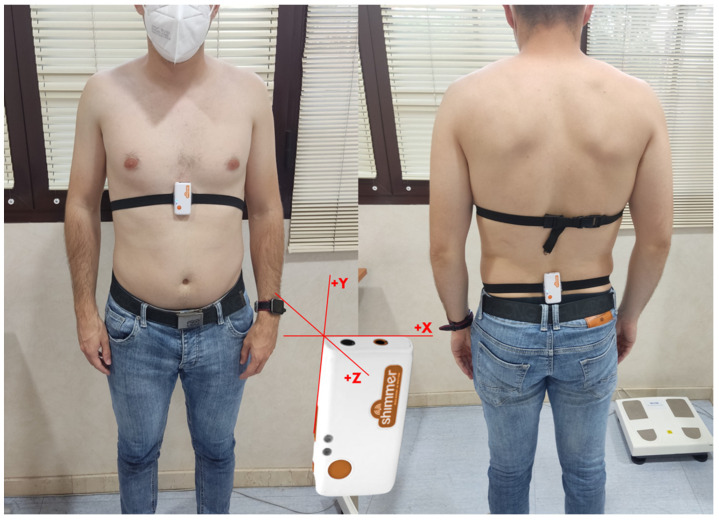
Shimmer 3 IMU: Placement and coordinate references (X, Y, Z axes).

**Table 1 jcm-12-00241-t001:** Lumbar and chest linear acceleration and angular velocity in the three dimensions during the 6MWT.

	Linear Acceleration (m/s^2^)				Angular Velocity (°/s)			
	Chest		Lumbar		Chest		Lumbar	
	Mean (SD)	Min–Max	Mean (SD)	Min–Max	Mean (SD)	Min–Max	Mean (SD)	Min–Max
**X Mean**	0.03 (0.78)	−2.56, 1.64	0.19 (0.63)	−1.57, 2.33	−0.68 (1.12)	−2.81, 1.77	0.003 (0.72)	−1.71, 2.48
**Y Mean**	9.15 (0.62)	7.13, 9.91	9.85 (0.66)	6.93, 10.42	0.71 (1.86)	−2.88, 7.12	0.46 (1.98)	−4.66, 8.46
**Z Mean**	3.28 (2.03)	−2.79, 7.10	1.65 (2.27)	−3.69, 6.80	−0.20 (0.89)	−2.41, 2.88	0.13 (0.68)	−2.36, 3.10
**X Max**	4.71 (2.67)	1.12, 18.19	6.08 (3.41)	1.02, 22.31	78.56 (70.78)	23.11, 492.50	124.33 (97.57)	30.78, 502.93
**Y Max**	14.52 (2.20)	10.10, 21.32	17.66 (3.29)	12.12, 24.59	135.76 (66.88)	35.59, 499.58	135.92 (56.11)	42.42, 314.83
**Z Max**	7.50 (3.64)	1.42, 24.60	7.49 (3.73)	−0.45, 21.55	68.76 (32.56)	23.38, 185.13	62.18 (26.49)	21.63, 143.57
**X Min**	−4.43 (2.23)	−14.47, −0.85	−6.14 (3.94)	−21.83, −1.53	−93.02 (55.25)	−326.53, −35.27	−101.71 (90.59)	−497.60, −26.22
**Y Min**	4.85 (5.61)	−38.82, 8.43	5.98 (2.19)	−6.25, 8.61	−127.27 (67.35)	−500.96, −31.62	−129.92 (69.70)	−497.89, −32.87
**Z Min**	−2.22 (4.61)	−23.73, 3.47	−3.01 (4.56)	−21.24, 3.83	−68.92 (32.33)	−187.17, −18.79	−61.66 (34.82)	−213.55, −20.99

**6MWT**: 6-Minute Walk Test; **SD**: Standard Deviation; **Max**: Maximum; **Min**: Minimum.

**Table 2 jcm-12-00241-t002:** Kinematic gait parameters differences between the ≥300 m and <300 m groups in the 6MWT and between patients who completed the 6MWT and patients who could not finish the 6MWT.

≥300 m (*n* = 22)															
	6MWT (m)	Lap Time (s)	Gait Speed (m/s)	Steps (n)	Step Time (s)	Step Length (m)	Step Velocity (m/s)	Step Cadence (Steps/Min)	Step Symmetry Ratio (Step Time Max/Step Time Min)	Strides (n)	Stride Time (s)	Stride Length (m)	Stride Velocity (m/s)	Stride Cadence (Strides/Min)	Stride Symmetry Ratio (Stride Time Max/Stride Time Min)
**Mean (SD)**	358.64 (60.16)	30.83 (4.56)	1.00 (0.17)	634.18 (65.18)	0.29 (0.09)	0.29 (0.08)	0.99 (0.16)	105.70 (10.86)	9.61 (14.60)	321.63 (48.53)	0.59 (0.19)	1.79 (0.45)	3.28 (1.05)	107.21 (16.18)	4.80 (5.79)
**<300 m (*n* = 48)**															
**Mean (SD)**	194.69 (50.42)	59.72 (17.52)	0.54 (0.14)	493.83 (90.88)	0.40 (0.11)	0.22 (0.07)	0.56 (0.14)	82.31 (15.15)	30.11 (25.26)	242.56 (53.16)	0.79 (0.22)	0.77 (0.30)	1.12 (0.62)	80.85 (17.72)	13.80 (10.78)
**≥300 m vs. <300 m difference**															
**Mean Difference** **(SE)** **(95%CI)**	163.95 *** (13.80) (136.40, 191.49)	−28.89 *** (2.71) (−34.32, −23.47)	0.46 *** (0.04) (0.38, 0.53)	140.35 *** (19.11) (102.06, 178.64)	−0.10 *** (0.03) (−0.16, −0.05)	0.07 *** (0.02) (0.03, 0.11)	0.43 *** (0.04) (0.36, 0.51)	23.39 *** (3.18) (17.01, 29.77)	−20.50 *** (4.79) (−30.08, −10.93)	79.07 *** (12.88) (53.11, 105.02)	−0.21 *** (0.05) (−0.31, −0.10)	1.02 *** (0.09) (0.83, 1.20)	2.17 *** (0.24) (1.68, 2.66)	26.36 *** (4.29) (17.70, 35.01)	−9.00 *** (1.99) (−12.96, −5.03)
**Completed the 6MWT (*n* = 41)**															
	**6MWT (m)**	**Lap Time (s)**	**Gait Speed (m/s)**	**Steps (n)**	**Step Time (s)**	**Step Length (m)**	**Step Velocity (m/s)**	**Step Cadence (Steps/Min)**	**Step Symmetry Ratio** **(Step Time Max/Step Time Min)**	**Strides (n)**	**Stride Time (s)**	**Stride Length (m)**	**Stride Velocity (m/s)**	**Stride Cadence (Steps/Min)**	**Stride Symmetry Ratio** **(Stride Time Max/Stride Time Min)**
**Mean (SD)**	289.76 (90.40)	41.44 (14.87)	0.80 (0.25)	596.07 (78.12)	0.33 (0.12)	0.26 (0.09)	0.81 (0.24)	99.35 (13.02)	9.88 (14.28)	292.15 (62.92)	0.66 (0.24)	1.36 (0.60)	2.37 (1.32)	97.38 (20.97)	4.65 (4.40)
**Stop in the 6MWT (*n* = 29)**															
**Mean (SD)**	184.66 (55.56)	63.65 (19.13)	0.51 (0.15)	455.76 (83.67)	0.41 (0.10)	0.21 (0.05)	0.53 (0.15)	75.96 (13.94)	43.16 (22.24)	232.44 (45.40)	0.82 (0.19)	0.71 (0.31)	0.99 (0.60)	77.48 (15.13)	19.92 (9.72)
**Completed the 6MWT vs. Stop in the 6MWT difference**															
**Mean Difference** **(SE)** **(95%CI)**	105.10 *** (17.49) (70.20, 140.01)	−22.21 *** (4.24) (−30.73, −13.69)	0.29 *** (0.05) (0.19, 0.39)	140.31 *** (19.52) (101.36, 179.27)	−0.08 ** (0.03) (−0.13, −0.02)	0.05 ** (0.02) (0.02, 0.08)	0.28 *** (0.05) (0.19, 0.37)	23.39 *** (3.25) (16.89, 29.88)	−33.27 *** (4.69) (−42.73, −23.81)	59.71 *** (13.68) (32.42, 87.00)	−0.015 ** (0.05) (−0.26, −0.05)	0.65 *** (0.11) (0.43, 0.87)	1.38 *** (0.23) (0.91, 1.85)	19.90 *** (4.56) (10.81, 29.00)	−15.27 *** (1.93) (−19.19, −11.36)

**6MWT**: Six-Minute Walk Test; **SD**: Standard Deviation; **SE**: Standard Error; **95% CI**: 95% Confidence Interval. ** *p* < 0.05; *** *p* < 0.001.

**Table 3 jcm-12-00241-t003:** Kinematic outcomes differences in the three dimensions between the ≥300 m and <300 m groups in the 6MWT.

Linear Acceleration (m/s^2^)									
Lumbar									
≥300 m (*n* = 22)									
	X Mean	Y Mean	Z Mean	X Max	Y Max	Z Max	X Min	Y Min	Z Min
**Mean (SD)**	0.13 (0.59)	9.98 (0.57)	1.31 (2.10)	8.57 (4.41)	20.50 (2.67)	8.94 (2.45)	−9.36 (4.94)	5.08 (2.84)	−3.91 (3.72)
**<300 m (*n* = 48)**									
**Mean (SD)**	0.23 (0.64)	9.79 (0.69)	1.80 (2.36)	4.93 (2.04)	16.35 (2.69)	6.82 (4.03)	−4.67 (2.20)	6.39 (1.70)	−2.60 (4.87)
**≥300 m vs. <300 m difference**									
**Mean Difference** **(SE)** **(95%CI)**	−0.10 (0.16) (−0.42, 0.22)	0.19 (0.17) (−0.15, 0.53)	−0.49 (0.59) (−1.66, 0.68)	3.64 *** (0.99) (1.62, 5.67)	4.14 *** (0.69) (2.76, 5.52)	2.12 ** (0.78) (0.55, 3.68)	−4.69 *** (1.10) (−6.95, −2.42)	−1.31 ** (0.55) (−2.40, −0.22)	−1.31 (1.17) (−3.65, 1.02)
**Chest**									
**≥300 m (*n* = 22)**									
	**X Mean**	**Y Mean**	**Z Mean**	**X Max**	**Y Max**	**Z Max**	**X Min**	**Y Min**	**Z Min**
**Mean (SD)**	0.10 (0.90)	9.19 (0.70)	2.99 (2.30)	5.77 (3.28)	15.87 (1.92)	7.65 (3.66)	−4.92 (1.92)	4.98 (1.68)	−3.48 (4.41)
**<300 m (*n* = 48)**									
**Mean (SD)**	−0.002 (0.72)	9.13 (0.58)	3.41 (1.90)	4.21 (2.20)	13.87 (2.04)	7.44 (3.67)	−4.20 (2.35)	4.79 (6.75)	−1.62 (4.62)
**≥300 m vs. <300 m difference**									
**Mean Difference** **(SE)** **(95%CI)**	0.10 (0.20) (−0.31, 0.50)	0.06 (0.16) (−0.26, 0.38)	−0.42 (0.53) (−1.48, 0.63)	1.56 ** (0.67) (0.21, 2.90)	2.01 *** (0.52) (0.97, 3.05)	0.21 (0.95) (−1.68, 2.11)	−0.73 (0.58) (−1.88, 0.42)	0.19 (1.46) (−2.74, 3.11)	−1.86 (1.18) (−4.22, 0.50)
**Angular Velocity (°/s)**									
**Lumbar**									
**≥300 m (*n* = 22)**									
	**X Mean**	**Y Mean**	**Z Mean**	**X Max**	**Y Max**	**Z Max**	**X Min**	**Y Min**	**Z Min**
**Mean (SD)**	−0.15 (0.86)	0.44 (2.89)	0.20 (1.00)	153.52 (108.29)	194.12 (53.95)	82.06 (25.27)	−136.44 (107.70)	−188.86 (89.11)	−86.31 (40.75)
**<300 m (*n* = 48)**									
**Mean (SD)**	0.07 (0.64)	0.47 (1.43)	0.09 (0.47)	110.96 (90.31)	109.25 (31.77)	53.06 (21.80)	−85.79 (77.73)	−102.91 (34.82)	−50.36 (24.96)
**≥300 m vs. <300 m difference**									
**Mean Difference** **(SE)** **(95%CI)**	−0.23 (0.18) (−0.60, −0.14)	−0.03 (0.65) (−1.37, 1.31)	0.10 (0.22) (−0.36, 0.56)	42.56 (24.77) (−6.88, 91.99)	84.87 *** (12.38) (59.50, 110.24)	28.99 *** (5.90) (17.22, 40.77)	−50.64 ** (22.68) (−95.90, −5.39)	−85.96 *** (19.65) (−126.52,−45.40)	−35.95 *** (9.41) (−55.20,−16.69)
**Chest**									
**≥300 m (*n* = 22)**									
	**X Mean**	**Y Mean**	**Z Mean**	**X Max**	**Y Max**	**Z Max**	**X Min**	**Y Min**	**Z Min**
**Mean (SD)**	−0.89 (1.29)	1.19 (2.72)	−0.24 (1.06)	93.03 (93.99)	181.75 (39.77)	79.48 (30.04)	−101.90 (51.33)	−156.46 (53.25)	−75.08 (27.01)
**<300 m (*n* = 48)**									
**Mean (SD)**	−0.57 (1.03)	0.49 (1.25)	−0.19 (0.81)	71.64 (56.43)	113.76 (66.27)	63.63 (32.77)	−88.77 (57.08)	−113.30 (69.39)	−65.97 (34.47)
**≥300 m vs. <300 m difference**									
**Mean Difference** **(SE)** **(95%CI)**	−0.32 (0.31) (−0.96, 0.32)	0.70 (0.61) (−0.55, 1.95)	−0.05 (0.23) (−0.51, 0.42)	21.39 (18.30) (−15.14, 57.92)	67.99 *** (12.94) (42.13, 93.85)	15.85 (8.28) (−0.68, 32.37)	−13.13 (14.34) (−41.76, 15.50)	−43.16 ** (16.77) (−76.64, −9.68)	−9.11 (8.37) (−25.82, 7.60)

**6MWT**: Six-Minute Walk Test; **SD**: Standard Deviation; **SE**: Standard Error; **95% CI**: 95% Confidence Interval. ** *p* < 0.05; *** *p* < 0.001.

**Table 4 jcm-12-00241-t004:** Kinematic outcomes differences in the three dimensions between patients who completed the 6MWT and patients who could not finish the 6MWT.

Linear Acceleration (m/s^2^)									
Lumbar									
Completed the 6MWT (*n* = 41)									
	X Mean	Y Mean	Z Mean	X Max	Y Max	Z Max	X Min	Y Min	Z Min
**Mean (SD)**	0.09 (0.67)	9.88 (0.56)	1.48 (2.26)	6.28 (3.06)	18.12 (3.36)	7.58 (3.29)	−6.59 (3.96)	6.10 (1.38)	−2.54 (3.36)
**Stop in the 6MWT (*n* = 29)**									
**Mean (SD)**	0.34 (0.53)	9.81 (0.79)	1.89 (2.32)	5.79 (3.90)	17.00 (3.14)	7.36 (4.33)	−5.51 (3.89)	5.81 (3.01)	−3.68 (5.86)
**Completed the 6MWT vs. Stop in the 6MWT difference**									
**Mean Difference** **(SE)** **(95%CI)**	−0.25 (0.15) (−0.55, 0.05)	0.07 (0.16) (−0.26, 0.39)	−0.42 (0.55) (−1.52, 0.69)	0.49 (0.83) (−1.17, 2.15)	1.12 (0.79) (−0.46, 2.71)	0.22 (0.91) (−1.60, 2.04)	−1.08 (0.95) (−2.98, 0.83)	0.29 (0.60) (−0.93, 1.50)	1.14 (1.21) (−1.30, 3.57)
**Chest**									
**Completed the 6MWT (*n* = 41)**									
	**X Mean**	**Y Mean**	**Z Mean**	**X Max**	**Y Max**	**Z Max**	**X Min**	**Y Min**	**Z Min**
**Mean (SD)**	0.02 (0.86)	9.25 (0.61)	2.94 (2.09)	4.80 (2.72)	14.96 (2.03)	6.73 (3.15)	−4.38 (1.79)	5.46 (1.55)	−2.28 (3.89)
**Stop in the 6MWT (*n* = 29)**									
**Mean (SD)**	0.04 (0.67)	9.00 (0.61)	3.76 (1.89)	4.59 (2.65)	13.88 (2.32)	8.62 (4.04)	−4.50 (2.78)	3.98 (8.56)	−2.14 (5.55)
**Completed the 6MWT vs. Stop in the 6MWT difference**									
**Mean Difference** **(SE)** **(95%CI)**	−0.01 (0.19) (−0.40, 0.37)	0.25 (0.15) (−0.05, 0.55)	−0.81 (0.49) (−1.80, 0.17)	0.22 (0.66) (−1.11, 1.54)	1.09 ** (0.53) (0.03, 2.15)	−1.89 ** (0.87) (−3.63, −0.15)	0.12 (0.55) (−0.99, 1.22)	1.48 (1.64) (−1.87, 4.83)	−0.14 (1.14) (−2.42, 2.15)
**Angular Velocity (°/s)**									
**Lumbar**									
**Completed the 6MWT (*n* = 41)**									
	**X Mean**	**Y Mean**	**Z Mean**	**X Max**	**Y Max**	**Z Max**	**X Min**	**Y Min**	**Z Min**
**Mean (SD)**	0.04 (0.86)	0.61 (2.39)	0.09 (0.78)	119.02 (87.69)	153.34 (60.00)	66.10 (27.25)	−100.48 (90.38)	−139.89 (58.17)	−62.69 (29.63)
**Stop in the 6MWT (*n* = 29)**									
**Mean (SD)**	−0.05 (0.46)	0.25 (1.20)	0.18 (0.50)	131.85 (111.25)	111.29 (39.29)	56.63 (24.77)	−103.45 (92.47)	−115.83 (82.38)	−60.20 (41.59)
**Completed the 6MWT vs. Stop in the 6MWT difference**									
**Mean Difference** **(SE)** **(95%CI)**	0.09 (0.16) (−0.23, 0.41)	0.36 (0.43) (−0.51, 1.23)	−0.09 (0.17) (−0.42, 0.24)	−12.84 (23.80) (−60.32, 34.65)	42.05 *** (11.88) (18.35, 65.75)	9.47 (6.37) (−3.24, 22.19)	2.98 (22.14) (−41.20, 47.15)	−24.05 (16.78) (−57.54, 9.44)	−2.49 *** (8.50) (−19.46, 14.48)
**Chest**									
**Completed the 6MWT (*n* = 41)**									
	**X Mean**	**Y Mean**	**Z Mean**	**X Max**	**Y Max**	**Z Max**	**X Min**	**Y Min**	**Z Min**
**Mean (SD)**	−0.69 (1.17)	0.99 (2.16)	−0.26 (0.96)	75.68 (72.42)	145.76 (49.98)	67.96 (28.32)	−90.69 (46.99)	−131.61 (51.71)	−71.84 (34.27)
**Stop in the 6MWT (*n* = 29)**									
**Mean (SD)**	−0.66 (1.06)	0.32 (1.28)	−0.13 (0.79)	82.67 (69.47)	121.47 (84.45)	69.90 (38.33)	−96.34 (66.07)	−121.06 (85.59)	−64.74 (29.43)
**Completed the 6MWT vs. Stop in the 6MWT difference**									
**Mean Difference** **(SE)** **(95%CI)**	−0.04 (0.28) (−0.59, 0.52)	0.66 (0.42) (−0.17, 1.50)	−0.13 (0.22) (−0.57, 0.31)	−6.99 (17.55) (−42.03, 28.06)	24.29 (16.33) (−8.32, 56.89)	−1.94 (8.08) (−18.07, 14.19)	5.65 (13.70) (−21.70, 33.00)	−10.55 (16.67) (−43.84, 22.73)	−7.10 (7.98) (−23.03, 8.83)

**6MWT**: Six-Minute Walk Test; **SD**: Standard Deviation; **SE**: Standard Error; **95% CI**: 95% Confidence Interval. ** *p* < 0.05; *** *p* < 0.001.

**Table 5 jcm-12-00241-t005:** Kinematic gait parameters differences between women and men and between patients who had fallen in the last year and those who did not fall in the last year in the 6MWT.

Women (*n* = 41)															
	6MWT (m)	Lap Time (s)	Gait Speed (m/s)	Steps (n)	Step Time (s)	Step Length (m)	Step Velocity (m/s)	Step Cadence (Steps/min)	Step Symmetry Ratio (Step Time Max/Step Time Min)	Strides (n)	Stride Time (s)	Stride Length (m)	Stride Velocity (m/s)	Stride Cadence (Strides/Min)	Stride Symmetry Ratio (Stride Time Max/Stride Time Min)
**Mean (SD)**	229.39 (75.62)	52.41 (17.94)	0.64 (0.21)	543.39 (98.62)	0.35 (0.09)	0.22 (0.07)	0.65 (0.20)	90.56 (16.44)	21.45 (19.47)	269.85 (57.67)	0.71 (0.18)	1.01 (0.49)	1.65 (1.09)	89.95 (19.22)	10.32 (8.67)
**Men (*n* = 29)**															
**Mean (SD)**	270 (110.88)	48.14 (22.59)	0.75 (0.31)	530.24 (116.88)	0.38 (0.15)	0.26 (0.09)	0.76 (0.29)	88.37 (19.48)	26.81 (29.97)	263.95 (71.46)	0.76 (0.29)	1.21 (0.70)	2.01 (1.49)	87.98 (23.82)	11.90 (12.44)
**Women vs. Men difference**															
**Mean Difference** **(SE)** **(95%CI)**	−40.61 (23.74) (−88.39, 7.17)	4.27 (4.85) (−5.41, 13.94)	−0.11 (0.07) (−0.25, 0.02)	13.15 (25.85) (−38.43, 64.72)	−0.03 (0.03) (−0.09, 0.04)	−0.04 ** (0.02) (−0.08, −0.01)	−0.12 (0.06) (−0.24, 0.01)	2.19 (4.31) (−6.40, 10.79)	−5.36 (6.34) (−18.14, 7.42)	5.90 (15.46) (−24.95, 36.75)	−0.05 (0.06) (−0.18, 0.07)	−0.20 (0.15) (−0.50, 0.11)	−0.37 (0.31) (−0.98, 0.25)	1.97 (5.15) (−8.32, 12.25)	−1.57 (2.68) (−6.96, 3.82)
**Fallen in the last year (*n* = 39)**															
	**6MWT (m)**	**Lap Time (s)**	**Gait Speed (m/s)**	**Steps (n)**	**Step Time (s)**	**Step Length (m)**	**Step Velocity (m/s)**	**Step Cadence (Steps/Min)**	**Step Symmetry Ratio** **(Step Time Max/Step Time Min)**	**Strides (n)**	**Stride Time (s)**	**Stride Length (m)**	**Stride Velocity (m/s)**	**Stride Cadence (Steps/Min)**	**Stride Symmetry Ratio** **(Stride Time Max/Stride Time Min)**
**Mean (SD)**	235 (80.29)	51.94 (19.58)	0.65 (0.22)	532.90 (102.83)	0.36 (0.11)	0.23 (0.07)	0.67 (0.21)	88.82 (17.14)	22.69 (24.61)	264.92 (60.61)	0.73 (0.22)	1.03 (0.51)	1.66 (1.09)	88.31 (20.20)	10.41 (10.38)
**Did Not Fall in the last year (*n* = 31)**															
**Mean (SD)**	260.32 (107.23)	49.01 (20.62)	0.72 (0.30)	544.29 (111.13)	0.36 (0.13)	0.25 (0.09)	0.73 (0.29)	90.72 (18.52)	24.90 (24.30)	270.55 (67.44)	0.73 (0.25)	1.17 (0.68)	1.96 (1.48)	90.18 (22.48)	11.69 (10.42)
**Fallen vs. Did Not Fall difference**															
**Mean Difference** **(SE)** **(95%CI)**	−25.32 (23.16) (−71.74, 21.10)	2.92 (4.82) (−6.70, 12.55)	−0.07 (0.06) (−0.20, 0.06)	−11.39 (25.64) (−62.56, 39.78)	0.001 (0.03) (−0.06, 0.06)	−0.02 (0.02) (−0.06, 0.02)	−0.07 (0.06) (−0.20, 0.06)	−1.90 (4.27) (−10.43, 6.63)	−2.20 (5.89) (−13.95, 9.55)	−5.63 (15.33) (−36.23, 24.96)	0.001 (0.06) (−0.11, 0.11)	−0.14 (0.15) (−0.44, 0.16)	−0.30 (0.32) (−0.94, 0.34)	−1.88 (5.11) (−12.08, 8.32)	−1.28 (2.50) (−6.27, 2.71)

**6MWT**: Six-Minute Walk Test; **SD**: Standard Deviation; **SE**: Standard Error; **95% CI**: 95% Confidence Interval. ** *p* < 0.05.

**Table 6 jcm-12-00241-t006:** Kinematic outcomes differences in the three dimensions between women and men in the 6MWT.

Linear Acceleration (m/s^2^)									
Lumbar									
Women (*n* = 41)									
	X Mean	Y Mean	Z Mean	X Max	Y Max	Z Max	X Min	Y Min	Z Min
**Mean (SD)**	0.07 (0.56)	9.79 (0.75)	1.58 (2.42)	5.26 (2.81)	16.88 (2.86)	6.81 (3.46)	−5.03 (2.32)	6.69 (1.40)	−2.53 (4.56)
**Men (*n* = 29)**									
**Mean (SD)**	0.37 (0.68)	9.94 (0.51)	1.75 (2.08)	7.23 (3.88)	18.75 (3.59)	8.44 (3.95)	−7.72 (5.11)	4.98 (2.70)	−3.69 (4.54)
**Women vs. Men difference**									
**Mean Difference** **(SE)** **(95%CI)**	−0.29 (0.15) (−0.59, 0.004)	−0.15 (0.16) (−0.47, 0.17)	−0.16 (0.56) (−1.27, 0.94)	−1.97 ** (0.80) (−3.57, −0.38)	−1.87 ** (0.77) (−3.41, −0.33)	−1.63 (0.89) (−3.40, 0.15)	2.69 ** (1.02) (0.63, 4.76)	1.71 *** (0.49) (0.73, 2.70)	1.16 (1.10) (−1.05, 3.36)
**Chest**									
**Women (*n* = 41)**									
	**X Mean**	**Y Mean**	**Z Mean**	**X Max**	**Y Max**	**Z Max**	**X Min**	**Y Min**	**Z Min**
**Mean (SD)**	−0.04 (0.81)	9.00 (0.69)	3.67 (2.06)	4.96 (3.23)	13.85 (1.96)	8.11 (4.18)	−4.62 (2.69)	4.17 (7.20)	−1.81 (5.61)
**Men (*n* = 29)**									
**Mean (SD)**	0.13 (0.74)	9.37 (0.42)	2.72 (1.90)	4.37 (1.56)	15.47 (2.21)	6.64 (2.51)	−4.16 (1.34)	5.84 (1.19)	−2.81 (2.59)
**Women vs. Men difference**									
**Mean Difference** **(SE)** **(95%CI)**	−0.17 (0.19) (−0.56, 0.21)	−0.37 ** (0.13) (−0.64, −0.10)	0.95 (0.49) (−0.04, 1.93)	0.59 (0.66) (−0.73, 1.91)	−1.62 ** (0.51) (−2.64, −0.61)	1.46 (0.88) (−0.30, 3.23)	−0.46 (0.49) (−1.45, 0.53)	−1.67 (1.38) (−4.42, 1.08)	1.00 (1.01) (−1.03, 3.03)
**Angular Velocity (°/s)**									
**Lumbar**									
**Women (*n* = 41)**									
	**X Mean**	**Y Mean**	**Z Mean**	**X Max**	**Y Max**	**Z Max**	**X Min**	**Y Min**	**Z Min**
**Mean (SD)**	−0.03 (0.82)	0.27 (1.43)	0.09 (0.48)	107.91 (101.23)	123.79 (48.30)	54.83 (22.12)	−92.43 (93.52)	−116.02 (45.62)	−52.63 (24.61)
**Men (*n* = 29)**									
**Mean (SD)**	0.05 (0.56)	0.73 (2.58)	0.18 (0.89)	147.55 (88.70)	153.07 (62.49)	72.56 (28.97)	−114.82 (86.17)	−149.58 (91.18)	−74.43 (42.83)
**Women vs. Men difference**									
**Mean Difference** **(SE)** **(95%CI)**	−0.08 (0.18) (−0.43, 0.27)	−0.46 (0.53) (−1.52, 0.61)	−0.10 (0.17) (−0.43, 0.23)	−39.64 (23.36) (−86.25, 6.97)	−29.28 ** (13.25) (−55.71, −2.84)	−17.73 ** (6.11) (−29.91, −5.54)	22.39 (21.98) (−21.46, 66.24)	33.56 (18.37) (−3.63, 70.75)	21.81 ** (8.83) (3.97, 39.64)
**Chest**									
**Women (*n* = 41)**									
	**X Mean**	**Y Mean**	**Z Mean**	**X Max**	**Y Max**	**Z Max**	**X Min**	**Y Min**	**Z Min**
**Mean (SD)**	−0.40 (1.03)	0.29 (1.27)	−0.29 (0.87)	87.63 (89.39)	129.14 (73.76)	72.72 (37.60)	−88.06 (57.78)	−128.66 (73.68)	−71.19 (35.81)
**Men (*n* = 29)**									
**Mean (SD)**	−1.08 (1.14)	1.32 (2.37)	−0.08 (0.92)	65.60 (24.32)	145.22 (55.53)	63.11 (23.07)	−110.09 (51.61)	−125.28 (58.37)	−65.67 (26.89)
**Women vs. Men difference**									
**Mean Difference** **(SE)** **(95%CI)**	0.68 ** (0.26) (0.15, 1.20)	−1.02 ** (0.49) (−2.02, −0.03)	−0.21 (0.22) (−0.65, 0.23)	22.03 (14.86) (−7.87, 51.93)	−16.08 (16.48) (−49.00, 16.83)	9.61 (7.37) (−5.11, 24.34)	12.03 (13.64) (−15.20, 39.25)	−3.38 (16.72) (−36.75, 30.00)	−5.52 (8.00) (−21.48, 10.45)

**6MWT**: Six-Minute Walk Test; **SD**: Standard Deviation; **SE**: Standard Error; **95%CI**: 95% Confidence Interval. ** *p* < 0.05; *** *p* < 0.001.

**Table 7 jcm-12-00241-t007:** Kinematic outcomes differences in the three dimensions between patients who had fallen in the last year and those who did not fall in the last year in the 6MWT.

Linear Acceleration (m/s^2^)									
Lumbar									
Fallen in the Last Year (*n* = 39)									
	X Mean	Y Mean	Z Mean	X Max	Y Max	Z Max	X Min	Y Min	Z Min
**Mean (SD)**	0.10 (0.52)	9.84 (0.73)	1.31 (2.41)	5.67 (2.70)	17.37 (3.26)	6.81 (3.46)	−5.90 (3.18)	6.59 (1.25)	−2.76 (3.67)
**Did Not Fall in the last year (*n* = 31)**									
**Mean (SD)**	0.31 (0.73)	9.87 (0.57)	2.08 (2.05)	6.58 (4.13)	18.02 (3.36)	8.34 (3.93)	−6.45 (4.77)	5.22 (2.83)	−3.33 (5.52)
**Fallen vs. Did Not Fall difference**									
**Mean Difference** **(SE)** **(95%CI)**	−0.21 (0.15) (−0.50, 0.09)	−0.03 (0.16) (−0.35, 0.29)	−0.77 (0.54) (−1.85, 0.32)	−0.91 (0.82) (−2.55, 0.73)	−0.65 (0.79) (−2.24, 0.93)	−1.53 (0.88) (−3.30, 0.23)	0.54 (0.95) (−1.36, 2.45)	1.36 ** (0.55) (0.26, 2.47)	0.57 (1.10) (−1.63, 2.77)
**Chest**									
**Fallen in the last year (*n* = 39)**									
	**X Mean**	**Y Mean**	**Z Mean**	**X Max**	**Y Max**	**Z Max**	**X Min**	**Y Min**	**Z Min**
**Mean (SD)**	0.06 (0.80)	9.15 (0.55)	3.29 (2.05)	5.05 (3.21)	14.31 (2.21)	7.66 (4.15)	−4.71 (2.65)	4.32 (7.42)	−2.62 (5.55)
**Did Not Fall in the last year (*n* = 31)**									
**Mean (SD)**	−0.005 (0.77)	9.15 (0.70)	3.26 (2.06)	4.29 (1.75)	14.78 (2.21)	7.31 (2.93)	−4.08 (1.52)	5.53 (1.29)	−1.72 (3.05)
**Fallen vs. Did Not Fall difference**									
**Mean Difference** **(SE)** **(95%CI)**	0.06 (0.19) (−0.32, 0.44)	−0.007 (0.15) (−0.31, 0.30)	0.03 (0.50) (−0.97, 1.03)	0.76 (0.65) (−0.55, 2.06)	−0.47 (0.54) (−1.54, 0.61)	0.35 (0.89) (−1.43, 2.14)	−0.64 (0.54) (−1.72, 0.45)	−1.21 (1.37) (−3.94, 1.53)	−0.90 (1.13) (−3.15, 1.35)
**Angular Velocity (°/s)**									
**Lumbar**									
**Fallen in the last year (*n* = 39)**									
	**X Mean**	**Y Mean**	**Z Mean**	**X Max**	**Y Max**	**Z Max**	**X Min**	**Y Min**	**Z Min**
**Mean (SD)**	0.009 (0.75)	0.36 (1.61)	−0.06 (0.53)	109.46 (63.76)	130.68 (48.33)	59.09 (22.98)	−86.87 (55.65)	−121.85 (49.81)	−56.24 (22.86)
**Did Not Fall in the last year (*n* = 31)**									
**Mean (SD)**	−0.004 (0.70)	0.59 (2.40)	0.36 (0.77)	143.04 (126.88)	142.52 (64.83)	66.06 (30.28)	−120.38 (119.61)	−140.07 (88.54)	−68.48 (45.17)
**Fallen vs. Did Not Fall difference**									
**Mean Difference** **(SE)** **(95%CI)**	0.01 (0.17) (−0.34, 0.36)	−0.23 (0.48) (−1.18, 0.73)	−0.43 ** (0.16) (−0.74, −0.12)	−33.58 (24.97) (−83.98, 16.81)	−11.84 (13.52) (−38.83, 15.15)	−6.97 (6.36) (−19.67, 5.73)	33.50 (23.26) (−13.49, 80.50)	18.22 (17.79) (−17.62, 54.05)	12.24 (8.90) (−5.72, 30.20)
**Chest**									
**Fallen in the last year (*n* = 39)**									
	**X Mean**	**Y Mean**	**Z Mean**	**X Max**	**Y Max**	**Z Max**	**X Min**	**Y Min**	**Z Min**
**Mean (SD)**	−0.55 (0.97)	0.62 (1.57)	−0.21 (0.89)	82.73 (90.66)	137.17 (74.77)	67.66 (32.29)	−98.46 (67.55)	−135.25 (77.98)	−68.58 (33.44)
**Did Not Fall in the last year (*n* = 31)**									
**Mean (SD)**	−0.84 (1.28)	0.84 (2.20)	−0.20 (0.91)	73.28 (32.18)	133.97 (56.53)	70.16 (33.40)	−86.11 (33.79)	−117.16 (50.33)	−69.35 (31.42)
**Fallen vs. Did Not Fall difference**									
**Mean Difference** **(SE)** **(95%CI)**	0.28 (0.27) (−0.26, 0.83)	−0.22 (0.46) (−1.13, 0.70)	−0.01 (0.22) (−0.45, 0.43)	9.44 (15.84) (−22.40, 41.28)	3.19 (16.45) (−29.66, 36.04)	−2.50 (8.01) (−18.48, 13.48)	−12.35 (12.58) (−37.53, 12.83)	−18.09 (16.42) (−50.88, 14.70)	0.77 (7.95) (−15.11, 16.66)

**6MWT**: Six-Minute Walk Test; **SD**: Standard Deviation; **SE**: Standard Error; **95%CI**: 95% Confidence Interval. ** *p* < 0.05.

**Table 8 jcm-12-00241-t008:** Discriminatory analysis of kinematic parameters in the 6MWT.

	AUC *
<300 m vs. >300 m	
All Subjects (*n* = 70)	
Step time	0.91
Step symmetry	0.88
Stride time	0.91
Stride symmetry	0.89
Lumbar Minimun Angular Velocity Y Axis	0.87
Lumbar Minimun Angular Velocity Z Axis	0.85
Chest Minimum Angular Velocity Y Axis	0.78
**Women (*n* = 41)**	
Step time	0.97
Step symmetry	0.95
Stride time	0.96
Stride symmetry	0.96
Lumbar Minimun Linear Acceleration X Axis	0.88
Lumbar Minimun Angular Velocity X Axis	0.83
Lumbar Minimun Angular Velocity Y Axis	0.95
Lumbar Minimun Angular Velocity Z Axis	0.99
Chest Minimum Angular Velocity Y Axis	0.85
**Men (*n* = 29)**	
Step time	0.90
Step symmetry	0.87
Stride time	0.90
Stride symmetry	0.86
Lumbar Minimun Linear Acceleration X Axis	0.76
Lumbar Minimun Angular Velocity Y Axis	0.79
Chest Minimum Angular Velocity Y Axis	0.76
**Patients who could not finish the 6MWT vs. Patients who finished the 6MWT**	
**All subjects (*n* = 70)**	
Step time	0.82
Step symmetry	0.95
Stride time	0.82
Stride symmetry	0.96
**Women (*n* = 41)**	
Step time	0.85
Step symmetry	0.99
Stride time	0.85
Stride symmetry	0.99
Lumbar Minimum Angular Velocity Y Axis	0.77
**Men (*n* = 29)**	
Step time	0.79
Step symmetry	0.92
Stride time	0.79
Stride symmetry	0.94
Lumbar Maximum Angular Velocity X Axis	0.76
**Women vs. Men**	
Lumbar Minimum Linear Acceleration Y Axis	0.76
**Fallen vs. Not Fallen**	
None	

6MWT: Six-Minute Walk Test; AUC: Area under the curve; * All the values had a significance level of *p* < 0.001.

## Data Availability

The data presented in this study are available on request from the corresponding author.

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
