# Peer review of "Kinematic Parameters That Can Discriminate in Levels of Functionality in the Six-Minute Walk Test in Patients with Heart Failure with a Preserved Ejection Fraction"

_jcm, 2022, doi:10.3390/jcm12010241_

Round 1

Reviewer 1 Report

Incorrect content:

line 48: "Thus, HF is the only cardiovascular disease increasing in incidence and prevalence due to the ageing of the world population"  atrial fibrillation incidence is also increasing due to ageing.

line 56: Approximately half of the patients with HF present a preserved left ventricular ejection fraction (HFpEF)  - Based on the Long Time Registry of the European Society of Cardiology the HFpEF is 16% 

Other:

The inclusion criteria must be more accurate regarding the patients with locomotive syndrome, it could destroy the results.

Fig1, is low quality, a figure with the axises marked their position on a schematic human body would be useful

Author Response

Point-by-point response to reviewers

Manuscript Number:   jcm-2028519

Relationship between quadriceps femoris muscle architecture and muscle strength and physical function in older adults with heart failure with preserved ejection fraction.

Dear Editors and referees,

Thank you very much for having in mind and review our manuscript. We are grateful for all the advice, comments and corrections suggested, which we believe are right to improve the quality of this paper.

Next, we will try to answer point by point to all your questions and suggestions, as well as specify the changes we have carried out.

Reviewer: 1

Incorrect content:

line 48: "Thus, HF is the only cardiovascular disease increasing in incidence and prevalence due to the ageing of the world population"  atrial fibrillation incidence is also increasing due to ageing.

Authors’ response: Thank you for your correction. We have corrected the sentence.

line 56: Approximately half of the patients with HF present a preserved left ventricular ejection fraction (HFpEF)  - Based on the Long Time Registry of the European Society of Cardiology the HFpEF is 16% 

Authors’ response: Thank you for your comment. We think that there is controversy about the data. Many studies indicate that approximately 50% of patients with HF present HFpEF(Bui et al., 2011; Dunlay et al., 2017; Groenewegen et al., 2020; Ho et al., 2016; McDonagh et al., 2021; Pfeffer et al., 2019). As you indicate, the Long Time Registry of the European Society of Cardiology showed that 60% of HF patients had HFrEF, 24% HFmrEF, and 16% HFpEF(Chioncel et al., 2017). Another study, however, determined that in hospitalized patients with HF, one study revealed a prevalence of HFrEF of 46%, HFmrEF of 8.2%, and HFpEF of 46%(Shah et al., 2017).

Bui, A. L., Horwich, T. B., & Fonarow, G. C. (2011). Epidemiology and risk profile of heart failure. Nature Reviews Cardiology, 8(1), 30–41. https://doi.org/10.1038/nrcardio.2010.165

Chioncel, O., Lainscak, M., Seferovic, P. M., Anker, S. D., Crespo-Leiro, M. G., Harjola, V. P., Parissis, J., Laroche, C., Piepoli, M. F., Fonseca, C., Mebazaa, A., Lund, L., Ambrosio, G. A., Coats, A. J., Ferrari, R., Ruschitzka, F., Maggioni, A. P., & Filippatos, G. (2017). Epidemiology and one-year outcomes in patients with chronic heart failure and preserved, mid-range and reduced ejection fraction: an analysis of the ESC Heart Failure Long-Term Registry. European Journal of Heart Failure, 19(12), 1574–1585. https://doi.org/10.1002/ejhf.813

Dunlay, S. M., Roger, V. L., & Redfield, M. M. (2017). Epidemiology of heart failure with preserved ejection fraction. Nature Reviews Cardiology, 14(10), 591–602. https://doi.org/10.1038/nrcardio.2017.65

Groenewegen, A., Rutten, F. H., Mosterd, A., & Hoes, A. W. (2020). Epidemiology of heart failure. European Journal of Heart Failure, 22(8), 1342–1356. https://doi.org/10.1002/ejhf.1858

Ho, J. E., Enserro, D., Brouwers, F. P., Kizer, J. R., Shah, S. J., Psaty, B. M., Bartz, T. M., Santhanakrishnan, R., Lee, D. S., Chan, C., Liu, K., Blaha, M. J., Hillege, H. L., van der Harst, P., van Gilst, W. H., Kop, W. J., Gansevoort, R. T., Vasan, R. S., Gardin, J. M., … Larson, M. G. (2016). Predicting Heart Failure With Preserved and Reduced Ejection Fraction: The International Collaboration on Heart Failure Subtypes. Circ Heart Fail, 9(6). https://doi.org/10.1161/CIRCHEARTFAILURE.115.003116 e003116

McDonagh, T. A., Metra, M., Adamo, M., Gardner, R. S., Baumbach, A., Böhm, M., Burri, H., Butler, J., Celutkiene, J., Chioncel, O., Cleland, J. G. F., Coats, A. J. S., Crespo-Leiro, M. G., Farmakis, D., Gilard, M., & Heymans, S. (2021). 2021 ESC Guidelines for the diagnosis and treatment of acute and chronic heart failure. European Heart Journal, 42(36), 3599–3726. https://doi.org/10.1093/eurheartj/ehab368

Pfeffer, M. A., Shah, A. M., & Borlaug, B. A. (2019). Heart Failure with Preserved Ejection Fraction in Perspective. Circulation Research, 124(11), 1598–1617. https://doi.org/10.1161/CIRCRESAHA.119.313572

Shah, K. S., Xu, H., Matsouaka, R. A., Bhatt, D. L., Heidenreich, P. A., Hernandez, A. F., Devore, A. D., Yancy, C. W., & Fonarow, G. C. (2017). Heart Failure With Preserved, Borderline, and Reduced Ejection Fraction: 5-Year Outcomes. Journal of the American College of Cardiology, 70(20), 2476–2486. https://doi.org/10.1016/j.jacc.2017.08.074

Other:

The inclusion criteria must be more accurate regarding the patients with locomotive syndrome, it could destroy the results.

Authors’ response: Thank you for your comment. In the research project we are carrying out, patients perform functional tests such as the 6MWT, the TUG, the SPPB and the 5-STS. Those who could not walk or stand up from a chair were excluded from the study. We think that if any patient had shown a severe locomotor syndrome, they would not have been able to walk or get up from a chair and would have been excluded. Regarding hospitalisation, we have made a mistake. We meant that patients hospitalised three months or less before participating in the study were excluded. With this exclusion criterion, we wanted to exclude those patients who had recently been hospitalised since this could affect their functionality. The eligibility criteria are intended to avoid bias from including a severely deconditioned sample. We have corrected the inclusion and exclusion criteria, improving their description. We think eligibility criteria can be better understood now.

Fig1, is low quality, a figure with the axises marked their position on a schematic human body would be useful

Authors’ response: Thank you very much for your advice. We have changed the figure 1.

We have also improved the writing in English. Additionally, we used the Grammarly software to check all the text in order to clarify some sentences and correct typos.

Reviewer 2 Report

In this work, authors challenge HEpEF patients to Six-Minute Walk Test (6MWT) to discriminate them according to their performance. Authors concluded that 6MWT can distinguish two different groups of HEpEF patients, and "gait kinematic outcomes could be useful to assess HFpEF patients during a different performance of 6MWT"

This is an interesting work, however I have some concerns:

1) Did authors find any difference related with gender difference? That could be at least discussed. In the same way, there is any difference between fallen and not fallen people in the parameters measured?

2) Is there difference in pharmacological treatments of HFpEF patients that could affect the results?

3) Could authors cross their data with other clinical outcomes, such VO2 or echocardiography data from these patients to validate their conclussion? In other way, is difficult to appreciate the relevance of the work. 

4) Please, units must be separated from numbers.

Author Response

Point-by-point response to reviewers

Manuscript Number:   jcm-2028519

Relationship between quadriceps femoris muscle architecture and muscle strength and physical function in older adults with heart failure with preserved ejection fraction.

Dear Editors and referees,

Thank you very much for having in mind and review our manuscript. We are grateful for all the advice, comments and corrections suggested, which we believe are right to improve the quality of this paper.

Next, we will try to answer point by point to all your questions and suggestions, as well as specify the changes we have carried out.

Reviewer: 2

In this work, authors challenge HEpEF patients to Six-Minute Walk Test (6MWT) to discriminate them according to their performance. Authors concluded that 6MWT can distinguish two different groups of HEpEF patients, and "gait kinematic outcomes could be useful to assess HFpEF patients during a different performance of 6MWT"

This is an interesting work, however I have some concerns:

Authors’ response: The authors sincerely appreciate the feedback, as well as all the comments and suggestions that have helped to improve the quality of the manuscript.

1) Did authors find any difference related with gender difference? That could be at least discussed. In the same way, there is any difference between fallen and not fallen people in the parameters measured?

Authors’ response: Thank you for your suggestion. We have now assessed the kinematics differences between women and men and the kinematics differences between fallen and not fallen people. We have included new tables and results. Moreover, we have changed the objectives as consequence of the new analyses.

2) Is there difference in pharmacological treatments of HFpEF patients that could affect the results?

Authors’ response: Thank you for your comment. As you can see in appendix B, the sample of patients with HFpEF were prescribed the usual drugs recommended in this population. We believe that the pharmacological treatment should not have affected the results.

3) Could authors cross their data with other clinical outcomes, such VO2 or echocardiography data from these patients to validate their conclussion? In other way, is difficult to appreciate the relevance of the work. 

Authors’ response: Thank you for your comment. Unfortunately, we do not have VO2 max data, and we have only included patients with HFpEF, so we cannot compare with HFrEF. Echocardiography data only includes left atrial dilation, left ventricular end-systolic dimension (LVESD), and left ventricular end-diastolic dimension (LVEDD). Thus, we cannot cross our results. The work highlights the use of kinematics as an objective evaluation of functionality (specifically, it assesses the biomechanical dimension) capable of discriminating between groups with different functional severity.

4) Please, units must be separated from numbers.

Authors’ response: Thank you for your correction. We have changed it.

Round 2

Reviewer 1 Report

The manuscript improved, suitable for publication.

Reviewer 2 Report

The authors have answered all of my questions